# Exploring Neural Granger Causality with xLSTMs: Unveiling Temporal Dependencies in Complex Data

**Harsh Poonia**[1,*]     **Felix Divo**[2,*]     **Kristian Kersting**[2,3,4,5]     **Devendra Singh Dhami**[6]

[1]Carnegie Mellon University     [2]AI & ML Group, TU Darmstadt     [3]hessian.AI
[4]DFKI     [5]Centre for Cognitive Science, TU Darmstadt     [6]TU Eindhoven

hpoonia@cs.cmu.edu     d.s.dhami@tue.nl
{felix.divo,kersting}@cs.tu-darmstadt.de

## Abstract

Causality in time series can be challenging to determine, especially in the presence of non-linear dependencies. Granger causality helps analyze potential relationships between variables, thereby offering a method to determine whether one time series can predict—Granger cause—future values of another. Although successful, Granger causal methods still struggle with capturing long-range relations between variables. To this end, we leverage the recently successful Extended Long Short-Term Memory (xLSTM) architecture and propose Granger causal xLSTMs (GC-xLSTM). It first enforces sparsity between the time series components by using a novel dynamic loss penalty on the initial projection. Specifically, we adaptively improve the model and identify sparsity candidates. Our joint optimization procedure then ensures that the Granger causal relations are recovered robustly. Our experimental evaluation on six diverse datasets demonstrates the overall efficacy of GC-xLSTM.

## 1   Introduction

Finding cause and effect among and within a group of multivariate time series can lead to a better understanding of the dynamics of the involved time series. For instance, in computational neuroscience and medicine, discovering brain connectivity assists in better understanding natural cognition [Smith et al., 2011]. Discovering inter-dependencies between time series also has a critical impact on many other research areas, such as finance [Masini et al., 2023], climate science [Mudelsee, 2019], and industrial applications [Strem et al., 2025]. Although efforts have recently been made to improve the interpretability of time series models [Ismail et al., 2020, Turbé et al., 2023], most methods are restricted to finding post-hoc interpretations and only focus on short-term dependencies.

The framework of Granger causality (GC) [Granger, 1969] was introduced to address the challenge of determining whether one variable's past values can help forecast another's future values, without implying direct causality. GC can be established using statistical hypothesis tests, determining whether one time series can predict another. The test traditionally involves estimating a vector autoregressive model and examining whether lagged values of a time series improve or degrade the prediction of the other, while controlling the past behavior of both series. Although GC does not imply a direct cause-and-effect relationship between the involved time series [Heckman, 2008], recognizing these interdependencies can lead to a better understanding of the dynamic relationships between variables over time [Marcinkevičs and Vogt, 2021, Shojaie and Fox, 2022].

---

*Authors contributed equally.

39th Conference on Neural Information Processing Systems (NeurIPS 2025).

Many families of deep learning architectures have been explored for time series analysis over the years, such as multilayer perceptrons [Zeng et al., 2023, Das et al., 2023], recurrent neural networks (RNNs) [Hochreiter and Schmidhuber, 1997, Cho et al., 2014], convolutional neural networks [Wu et al., 2022, Wang et al., 2022], Transformers [Vaswani et al., 2017, Nie et al., 2023], state-space models (SSMs) [Wang et al., 2025], or mixing architectures [Wang et al., 2024]. Throughout this, recurrent models remained a natural choice for time series data since their direction of computation aligns well with the forward flow of time. This aligns particularly well with the goals of neural Granger causality. Although SSMs are comparable in that regard, RNNs tend to offer more powerful forecasting capabilities since they deteriorate less when modeling long-term dependencies. Furthermore, their inference runtime is typically linear in the sequence length at constant memory cost, making them much more efficient than, for instance, Transformers with quadratic runtimes and memory requirements, while remaining highly expressive. Recently, Beck et al. [2024] revisited recurrent models by borrowing insights gained from Transformers in many domains, specifically natural language processing. Their proposed Extended Long Short-Term Memory (xLSTM) model sparked a resurgence of interest in recurrent architectures for sequence modeling and has already proven highly suitable for time series forecasting [Kraus et al., 2025, Alharthi and Mahmood, 2024].

Although most Granger causal machine learning methods assume linearity in time series as a fundamental prerequisite [Siggiridou and Kugiumtzis, 2015, Zhang et al., 2020], recent efforts capture non-linear dynamics in time series by using neural networks as the modeling choice instead of VARs [Tank et al., 2022, Löwe et al., 2022, Cheng et al., 2024]. Although successful, these non-linear methods require careful feature engineering to include time-based patterns. Thus, they may not capture interactions between time series and external factors as effectively as xLSTMs, which can learn non-linear patterns and adapt to the non-stationary nature of time series data.

We introduce *GC-xLSTM*, a novel method that leverages xLSTMs to uncover the GC relations in the presence of complex data, which inherently can have long-range dependencies.[1] GC-xLSTM first enforces sparsity between the time series components by using a novel lasso penalty on the initial projection layer of the xLSTM. We learn a weight per time series and then adapt them to find the relevant variates for that step. Then, each time series component is modeled using a separate xLSTM model, enabling us to discover interpretable GC relationships between the time series variables. After the forecast results by the individual xLSTM models, the important features are made more prominent, whereas the less important ones are diminished by a joint optimization technique, which includes using a novel reduction coefficient. Thus, the overall GC-xLSTM model can be trained end-to-end to uncover long-range Granger causal relations.

Our main research **contributions** can be summarized as follows:

(i) We propose GC-xLSTM, a novel model that can uncover Granger causal relations in non-linear time series.

(ii) Our novel algorithm jointly improves the forecasting model while adaptively enforcing strict sparsity.

(iii) Our empirical evaluations demonstrate that GC-xLSTM can robustly discover Granger causal relations in the presence of complex simulated and real-world data.

**Outline.** We start by recalling preliminaries and reviewing related research to contextualize this work in the broader body of research on neural Granger causality in Section 2. This allows us to introduce GC-xLSTM in Section 3 and empirically evaluate in relation to other methods in Section 4. Finally, we conclude with an outlook to future work in Section 5.

## 2 Preliminaries and Related Work

We are interested in datasets of strictly stationary time series $\boldsymbol{S} \in \mathbb{R}^{V \times T}$ of $V$ variates with length $T$. Let $\boldsymbol{S}_t \in \mathbb{R}^V$ denote the value of $\boldsymbol{S}$ at time $t$. A variate $v$ (sometimes called a channel) can be any scalar measurement, such as the chlorophyll content of a plant or the spatial location of some object being tracked. Its value at time $t$ is $S_{v,t} \in \mathbb{R}$. The measurements are assumed to be carried out jointly at $T$ regularly spaced time steps. In forecasting, a model is presented with a time series of $C$ context steps $\boldsymbol{S}_{<t}$ before $t$, from which it shall predict the next value $\boldsymbol{S}_t \in \mathbb{R}^V$.

---

[1]Code available at github.com/harpoonix/GC-xLSTM.

## 2.1 Granger Causality

If the observed time series were generated by some underlying process $g$, which we can formalize as a structural equation model for all time steps $t$ as $S_{v,t} = g_v(\boldsymbol{S}_{1,<t}, \ldots, \boldsymbol{S}_{V,<t}) + \epsilon_{v,t}$ for all $v \in \mathcal{V}$, where $\epsilon_{v,t}$ is some additive zero mean noise independent from all variates $\boldsymbol{S}_{v,<t}$ and $\mathcal{V} := \{1, \ldots, V\}$ is the set of all variates. In Granger causality [Granger, 1969], we aim to determine whether past values $\boldsymbol{S}_{v,<t}$ of a variate $v$ are predictive for future values $\boldsymbol{S}_{w,\geq t}$ of another variate $w$. Following the notation of Shojaie and Fox [2022], we formally define:

**Definition 1** (Granger Causality). *Variate $v$ is* Granger non-causal *for $w$ if and only if $g_v$ is invariant to $\boldsymbol{S}_{w,<t}$ for all $t \in \{1, \ldots, T\}$, i.e., if and only if*

$$g_v(\boldsymbol{S}_{1,<t}, \ldots, \boldsymbol{S}_{V,<t}) = g_v(\boldsymbol{S}_{1,<t}, \ldots, \boldsymbol{S}_{V,<t} \setminus \boldsymbol{S}_{w,<t}).$$

*Else, we call $v$* Granger causal *for $w$.*

The set of all such relationships are the directed edges $\mathcal{E} \subseteq \mathcal{V} \times \mathcal{V}$ of the *Granger causal graph* $(\mathcal{V}, \mathcal{E})$ of the variates, which we eventually aim to uncover.

## 2.2 Neural Granger Causality

Unfortunately, however, we cannot explicitly access $g$ in most realistic settings. Using machine learning methods, we can nonetheless estimate each of the $V$ process components $g_v$ by an autoregressive time series forecasting model $\mathcal{M}_{\boldsymbol{\theta},v}(\boldsymbol{S}_{<t}) \approx g_v(\boldsymbol{S}_{<t}) = \boldsymbol{S}_{v,t} - \epsilon_{v,t}$. We can do so by estimating the parameters $\boldsymbol{\theta}$ of the model based on the dataset of time series, minimizing the predictive mean squared error (MSE) loss

$$\mathcal{L}_{\text{pred}}(\boldsymbol{\theta}) = \sum_{v=1}^{V} \sum_{t=1}^{T} (S_{v,t} - \mathcal{M}_{\boldsymbol{\theta},v}(\boldsymbol{S}_{<t}))^2. \tag{1}$$

In the case of using neural networks for $\mathcal{M}_{\boldsymbol{\theta},v}$, this approach is called *Neural Granger Causality* [Tank et al., 2022].

It would be very costly to train a total of $V^2$ models to test if each variate Granger causes any other variate and thus construct the entire Granger causal graph. To avoid this, we can train merely a single *component-wise* model $\mathcal{M}_{\boldsymbol{\theta},v}$ for each variate $v$ and inspect what inputs $w$ it is sensitive to. While this does not ablate models as in classical GC, it reduces the number of models to be trained from quadratic to linear in $V$. This can, for instance, be achieved by optimizing the predictive loss $\mathcal{L}_{\text{pred}}(\boldsymbol{\theta})$ based on all model parameters $\boldsymbol{\theta}$ with a regularizer $\Omega(\widehat{\boldsymbol{\theta}}_v)$ enforcing sparsity in the input features:

$$\min_{\boldsymbol{\theta}, \widehat{\boldsymbol{\theta}}} \mathcal{L}_{\text{pred}}(\boldsymbol{\theta}) + \lambda \sum_{v=1}^{V} \Omega\left(\widehat{\boldsymbol{\theta}}_v\right), \tag{2}$$

where $\widehat{\boldsymbol{\theta}}_v$ are tunable parameters of the regularizer for variate $v$ and $\lambda \in \mathbb{R}_+$ is a hyperparameter to adjust the degree of sparsity. One such approach are cMLPs [Tank et al., 2022]; multilayer perceptrons where the first weight matrix $\boldsymbol{W} \in \mathbb{R}^{D \times V}$ projecting from $V$ features at each time step to $D$ hidden dimensions is regularized to encode a sparse selection of input features. $\Omega$ is instantiated as an $L^2$ norm of its columns: $\sum_{v=1}^{V} \|\boldsymbol{W}_v\|_2$. Note that $\boldsymbol{\theta}$ and $\widehat{\boldsymbol{\theta}}$ can overlap.

Sparsity can then be extracted by binarizing the entries of $\boldsymbol{W}$ using a user-defined threshold $\tau$. This is necessary as the $L^2$ penalty tends only to shrink parameters to small values near zero, yet not clamp them sharply to it. This, however, allows subsequent layers to amplify the dampened signal again and still use it for forecasting. We avoid this disadvantage in GC-xLSTM by explicitly optimizing the feature extractor for strict sparsity. This more principled approach works without determining a sparsity threshold $\tau$.

Previous work has explored both more regularizers [Tank et al., 2022] and different means to extract Granger causal relationships, including using feature attribution via explainability [Atashgahi et al., 2024] and interpretability [Marcinkevičs and Vogt, 2021]. Furthermore, several works have gone towards learning relevant representations that respect the underlying Granger causality [Xu et al., 2016, Varando et al., 2021, Dmochowski, 2023]. Zoroddu et al. [2024] present another approach where prior knowledge is encoded in the form of a noisy undirected graph, which aids the learning of

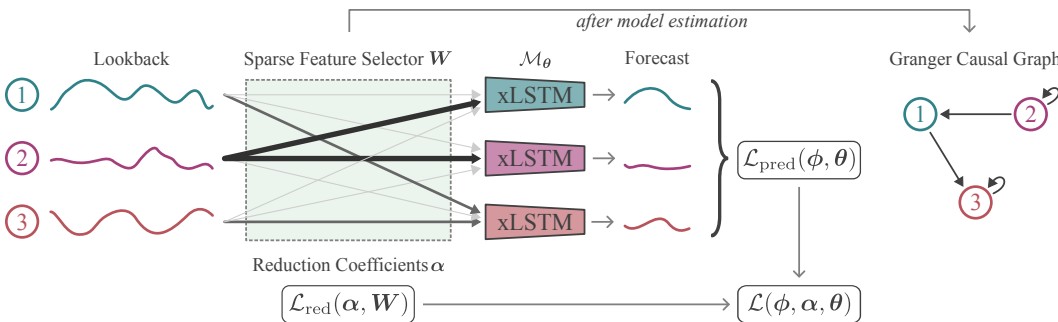

Figure 1: **GC-xLSTM performs three key steps to determine the Granger causal links:** Firstly, for each time series component, all variates are embedded with a sparse feature encoder $\boldsymbol{W}$ that is regularized through a novel sparsity loss with learned reduction coefficients $\boldsymbol{\alpha}$. xLSTM models then learn to autoregressively predict future steps from that embedding. Finally, once model estimation is complete, Granger causal dependencies can be extracted from $\boldsymbol{W}$.

Granger causality graphs. A more recent approach [Lin et al., 2024] employs Kolmogorov-Arnold networks [Liu et al., 2024] to learn the Granger causal relations between time series. For a discussion of when Granger causality implies (Pearlian) causality, we refer to Pettenuzzo and White [2011] and Das and Babadi [2023], allowing to connect to works such as the one of Rubenstein et al. [2017].

## 2.3 Extended Long Short-Term Memory (xLSTM)

Beck et al. [2024] propose two building blocks to build up xLSTM architectures: the sLSTM and mLSTM modules for vector-valued (i.e., multivariate) sequences. sLSTM cells improve upon classic LSTMs by exponential gating. For parallelizable training, mLSTM cells replace memory mixing between hidden states with an associative matrix memory. We will continue by recalling how sLSTM cells function since we found their memory mixing more effective in time series forecasting.

The standard LSTM architecture of Hochreiter and Schmidhuber [1997] updates the cell state $\mathbf{c}_t$ through a combination of input, forget, and output gates, which regulate the flow of information across tokens. sLSTM blocks, owing to the contained sLSTM cells, enhance this by incorporating exponential gating and memory mixing [Greff et al., 2017] to better handle complex temporal and cross-variate dependencies. Additional normalization states are introduced to stabilize training under the new exponential activation function. As Beck et al. have shown, it is sufficient and computationally beneficial to constrain the memory mixing performed by the recurrent weight matrices $\boldsymbol{R}_z, \boldsymbol{R}_i, \boldsymbol{R}_f$, and $\boldsymbol{R}_o$ to individual *heads*. This is inspired by the multi-head setup of Transformers [Vaswani et al., 2017], yet more restricted and efficient. In particular, each token gets broken up into groups of features, where the input weights $\boldsymbol{W}_{z,i,f,o}$ act across all of them, but the recurrence matrices $\boldsymbol{R}_{z,i,f,o}$ are implemented as block-diagonal. This permits specialization of the individual heads to patterns specific to the respective section of the tokens and empirically does not sacrifice expressivity.

# 3 GC-xLSTM

We will now introduce the GC-xLSTM architecture and detail the optimization for strict sparsity jointly with the model parameters. At the end, we will additionally discuss theoretical properties of the proposed system.

## 3.1 Overall Architecture

As Figure 1 shows, we estimate a pipeline of sparse feature selectors and xLSTM models to predict the multivariate time series. Eventually, this allows us to derive Granger causal dependencies from the selected features. Specifically, we learn for each variate $v$ a separate sparse projection and compute $\boldsymbol{x}_v = \boldsymbol{W}_v \boldsymbol{S} + \boldsymbol{b}_v$ shared across time. The matrix $\boldsymbol{W}_v \in \mathbb{R}^{D \times V}$ is shared across all lags of the time series for simplicity. We will lift this restriction in Section 4.2. Note that $\boldsymbol{b}$ does not affect the sparse use of inputs. We write $\boldsymbol{\phi}$ for the set of parameters $\boldsymbol{W}_v$ and $\boldsymbol{b}_v$ for all $v \in \mathcal{V}$. In addition to selecting dependencies, the sparse projection embeds the data into $D$-dimensional hidden space.

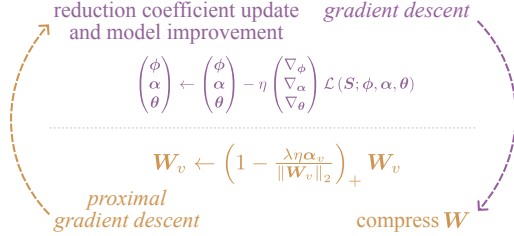
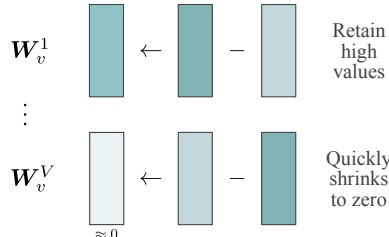

(a) **The optimization procedure for GC-xLSTM alternates between GD and proximal GD.** A high value of $\alpha_v^w$ entails a strong compression of column $W_v$, i.e. more sparsity in depending on variate $w$.

(b) **Visual representation of the compression step in Algorithm 1, line 16.** The matrix preserves columns with high norm that are important for Granger causality detection (dark) while suppressing others (light).

Figure 2: **Optimization procedure and compression intuition for GC-xLSTM.**

For Neural Granger Causality to successfully and faithfully extract the proper underlying dependencies, it is essential to employ models that can capture the complete set of dependencies. We, therefore, employ powerful deep-learning models with significantly higher capacity than the cMLPs and cLSTMs in prior work [Tank et al., 2022]. In particular, we instantiate the individual time series forecasters $\mathcal{M}_{\theta_v,v}$ with sLSTM blocks as introduced in Section 2.3. They can capture long-range dependencies in time series data, substantially enhancing the capabilities of traditional LSTMs in handling extended contexts. GC-xLSTM consists of $V$ sLSTM models, each modeling a different time series component. They are trained using established forecasting losses, such as the MSE loss $\mathcal{L}_{\text{pred}}(\phi, \theta)$ from Eq. (1).

## 3.2 Optimizing for Strict Input Sparsity

The purpose of the feature selector $W$ is to only "pay attention" to as many variates as necessary for successful forecasting. A common approach to achieve this sparsity on the Granger causal relationships is via the lasso regularization explained in Section 2.2. We use a variation of the group lasso penalty [Yuan and Lin, 2006, Simon and Tibshirani, 2012] on the initial projection layer of GC-xLSTM as a structured sparsity-inducing penalty that encourages the selection of entire groups of variables, encoded as the columns of $W_v$. Note that our penalty differs from adaptive lasso [Yuan and Lin, 2006], where the weights are not learned but are treated as fixed heuristics. Note that standard gradient descent methods cannot optimize such a penalty directly due to its non-differentiability. We thus adaptively compress by learning a reduction coefficient $\alpha_v \in \mathbb{R}^V$ that selects which of the $V$ columns of $W_v$ are redundant. We perform this compression of $W_v$ in a joint procedure with the general optimization of the forecasting model, as provided in Algorithm 1. Specifically, we perform two updates per optimization step for each of the variates, as Figure 2a depicts. Firstly, we optimize the projection weights $\phi$, the reduction coefficients $\alpha$, and the xLSTM parameters $\theta$ using mini-batch gradient descent. This corresponds to lines 10 to 15 in Algorithm 1. It optimizes the following loss expected over the time series data $S$:

$$\min_{\phi_v, \alpha_v, \theta_v} \mathcal{L}_{\text{pred}}(S; \phi_v, \theta_v) + \lambda \underbrace{\log\left(\sum_{w=1}^V \alpha_v^w \left\| \text{sg}\left(W_v^w\right)\right\|_2\right)}_{\mathcal{L}_{\text{red}}(\alpha_v, W_v)} \tag{3}$$

Note that we, crucially, only descend on the reduction coefficient $\alpha$ in $\mathcal{L}_{\text{red}}$ and not on $W_v$, as the stop-gradient $\text{sg}(\cdot)$ denotes. This sparsity optimization is instead performed by the second step in the procedure, shown in line 16, where a proximal gradient descent step dynamically shrinks $W_v$ proportional to $\alpha_v$. The compression update takes a descent step towards the gradient of

$$\lambda \sum_{w=1}^V \alpha_v^w \left\| W_v^w \right\|_2 \tag{4}$$

followed by a soft thresholding. Intuitively, the $\mathcal{L}_{\text{red}}$ component of Eq. (3) keeps $W_v$ fixed while learning $\alpha_v$, and Eq. (4) keeps $\alpha_v$ fixed in the proximal step to compress $W_v$. Figure 2b depicts the intuition of the proximal gradient step and soft-thresholding in line 16.

**Details on learning the reduction loss $\mathcal{L}_{\mathbf{red}}$.** It is worth briefly discussing the use of the logarithm in Eq. (3). It mainly gives more equal weight to the decreases in $\boldsymbol{W}_v$ column norms and encourages learning of better sparse Granger causal relations. It furthermore normalizes the gradient updates to $\boldsymbol{\alpha}_v$. Empirically, this loss engineering allowed training models that were significantly more robust to noise and changes to the sparsity hyperparameter $\lambda$. This was reflected by a more stable variable usage and predictive loss $\mathcal{L}_{\mathrm{pred}}$.

**Ensuring non-negativity of the reduction coefficients $\boldsymbol{\alpha}$.** For the proximal update step to be well-behaved, we need to ensure that $\boldsymbol{\alpha}_v$ results in a convex combination of column weights, i.e., that $\boldsymbol{\alpha}_v^w > 0$ for all $w \in \mathcal{V}$ and $\boldsymbol{\alpha}_v^T \mathbb{1} = 1$. We achieve this by re-parameterizing it as $\boldsymbol{\alpha}_v = \mathrm{softmax}\left(\boldsymbol{\beta}_v\right)$, and learning $\boldsymbol{\beta}_v$ instead of $\boldsymbol{\alpha}_v$.

**Intuitive dynamics of the gradient update step.** The weights $\boldsymbol{W}_v$ in the reduction loss $\mathcal{L}_{\mathrm{red}}$ of Eq. (3) only serve to learn good reduction coefficients $\boldsymbol{\alpha}_v$, and are not optimized themselves. Deriving the gradient of the penalty term $\mathcal{L}_{\mathrm{reg}}$ with respect to the underlying $\boldsymbol{\beta}$ provides a helpful intuition of the training dynamics:

$$\frac{\partial}{\partial \beta_v^w} \sum_{w=1}^{V} \alpha_v^w \left\| \boldsymbol{W}_v^w \right\|_2 = \frac{\partial}{\partial \beta_v^w} \sum_{w=1}^{V} \mathrm{softmax}(\boldsymbol{\beta}_v)^w \left\| \boldsymbol{W}_v^w \right\|_2 = \alpha_v^w \left( \left\| \boldsymbol{W}_v^w \right\|_2 - \sum_{w=1}^{V} \alpha_v^w \left\| \boldsymbol{W}_v^w \right\|_2 \right)$$

$$\implies \frac{\partial}{\partial \beta_v^w} \mathcal{L}_{\mathrm{reg}} = \lambda \frac{\partial}{\partial \beta_v^w} \log \left( \sum_{w=1}^{V} \alpha_v^w \left\| \boldsymbol{W}_v^w \right\|_2 \right) = \lambda \alpha_v^w \left( \frac{\left\| \boldsymbol{W}_v^w \right\|_2}{\sum_{w=1}^{V} \alpha_v^w \left\| \boldsymbol{W}_v^w \right\|_2} - 1 \right)$$

We can see that if the norm of a column $\left\| \boldsymbol{W}_v^w \right\|_2$ is large, that corresponding $\frac{\partial}{\partial \beta_v^w} \mathcal{L}_{\mathrm{reg}}$ will be large. Gradient descent will thus decrease $\alpha_v^w$ and effectively allocate less weight to its removal in the compression step. Furthermore, $\frac{\partial}{\partial \beta_v^w} \mathcal{L}_{\mathrm{reg}}$ also scales with $\alpha_v^w$, resulting in a self-reinforcing loop that aids learning sparse representations.

**Practical considerations.** Furthermore, we perform staged optimization of $\boldsymbol{\alpha}$, which is initialized to a uniform distribution by setting all $\boldsymbol{\beta} = 0$. We only start training $\boldsymbol{\alpha}$ after exploring the prediction loss and having obtained a reasonably compressed forecaster, which is controlled by the hyperparameter $K$ in Algorithm 1 (see line 13). While we present the method with mini-batch gradient descent for conciseness, modern optimizers, such as Adam [Kingma and Ba, 2017], can further improve convergence.

### 3.3 Theoretical Analysis

Ultimately, we want to ensure that, provided real-world data, we can find hyperparameters such that Algorithm 1 discovers all and only those edges of the unique underlying GC graph $(\mathcal{V}, \mathcal{E})$ as per Definition 1. Providing convergence guarantees in full generality is notoriously hard for such practical architectures and optimization schemes, and thus rarely attempted. However, we can at least investigate whether the chosen model class containing $\mathcal{M}_{\boldsymbol{\theta},v}$ can approximate $g_v$ to arbitrary precision. If that is the case, we can be reasonably sure that gradient-based optimization schemes will yield satisfactory approximations, even without formal guarantees.

The forecasting component of GC-xLSTM consists of two main steps: the sparse initial projection $\boldsymbol{W}_v$ and the subsequent xLSTM blocks. One might think that the sparsity of $\boldsymbol{W}_v$ hinders learning the correct $g_v$. However, the true underlying $g_v$ is independent of all variates $w$ without ingoing edges into variate $v$. Thus, depending on an appropriate choice of the sparsity hyperparameter $\lambda$, the projection $\boldsymbol{W}_v$ can encode exactly those as zero entries. It remains to investigate the approximation capabilities of the sLSTM blocks, which we present in Appendix B in more detail. In summary, we can assume that sLSTM blocks are at least as powerful as RNNs, which are, in turn, universal function approximators. Thus, the overall GC-xLSTM architecture is sufficiently rich to model $g_v$ to adequate precision. We continue by confirming this empirically in the next section.

Table 1: **GC-xLSTM is highly accurate at discovering GC relations in the chaotic and non-linear Lorentz-96 system.** For each setting and baseline, we provide the accuracy (Acc.), balanced accuracy (BA), and AUROC. The best models are highlighted as **bold**.

| Model | Year | $F = 10$ | | | $F = 40$ | | |
|---|---|---|---|---|---|---|---|
| | | Acc. (↑) | BA (↑) | AUROC (↑) | Acc. (↑) | BA (↑) | AUROC (↑) |
| VAR | 2021 | 91.8±1.2 | 83.8±1.6 | 94.0±1.6 | 86.4±0.8 | 58.5±1.7 | 74.5±4.7 |
| cLSTM | 2022 | 97.0±1.0 | 95.0±2.8 | 95.8±2.6 | 84.4±1.2 | 65.6±3.7 | 66.1±3.8 |
| cMLP | 2022 | 97.2±0.5 | 95.6±1.6 | 96.3±1.8 | 68.3±2.7 | 80.5±1.7 | **97.9**±1.6 |
| GC-KAN | 2024 | – | – | 92.1±0.3 | – | – | 87.1±0.4 |
| TCDF | 2019 | 87.1±1.2 | 70.9±4.4 | 85.7±2.7 | 77.5±2.3 | 62.2±3.0 | 67.9±3.1 |
| eSRU | 2020 | 96.6±1.1 | 95.1±2.0 | 96.3±2.0 | 86.7±0.9 | 88.6±1.4 | 93.4±2.1 |
| GVAR | 2021 | 98.2±0.3 | 98.2±0.6 | **99.7**±0.1 | 94.5±1.0 | 88.5±4.6 | 97.0±0.9 |
| **GC-xLSTM** | **ours** | **99.1**±0.2 | **98.5**±1.0 | 99.3±0.3 | **96.3**±0.3 | **96.6**±0.3 | 88.0±0.2 |

## 4 Experimental Evaluation

We conduct extensive experiments on six datasets to assess the practical effectiveness of GC-xLSTM. We will now explain the chosen architecture and parameters used to train GC-xLSTM and then discuss the datasets used before presenting our obtained results in detail. The results are split into investigating the general GC modeling capabilities on a diverse range of applications (Section 4.1) and a subsequent model analysis, including an ablation study and using different numbers of variates (Section 4.2).

**Architecture Details.** For our component-wise networks, we use a single xLSTM block comprised of one sLSTM layer, followed by a linear layer to predict the next time step from the preceding $C = 10$ time steps (see Table 5 in Appendix C for deviations). We then directly optimize the architecture on each time step. The hidden dimension of the sLSTM block is set to 32 for all datasets. We find that the presence of a gated MLP to up- and down-project the hidden states of the sLSTM block does not significantly improve performance, so we omit it in all our experiments for simplicity. We deliberately do not use any mLSTM blocks, as we also find that the sLSTM blocks are superior at capturing long-range dependencies in the data [Kraus et al., 2025].

**Training and Evaluation Details.** We use the Adam optimizer [Kingma and Ba, 2017] for full gradient descent training with a weight decay of 0.1. We schedule the learning rate to follow a linear warmup of 2,000 iterations to $\eta = 10^{-4}$, followed by cosine annealing until the end of training for a total of 13,000 steps. We start learning the reduction coefficients after a warmup period of $K = 1,500$ iterations, during which the uniform compression across all columns combined with the prediction loss gives reasonable priors for the gradient directions of the reduction coefficients. Due to the moderate size of the datasets, we performed full-batch gradient descent. Only the sparsity hyperparameter $\lambda$ was tuned specifically for each setting. This allows obtaining degrees of sparsity specifically tailored to the characteristics and requirements of each dataset and task. Note that, except for the customization of $\lambda$, we use essentially the same hyperparameter configuration for an extensive set of datasets, underpinning the robustness of GC-xLSTM. Following [Tank et al., 2022, Sec. 6.1], we compute all AUROC scores by sweeping over $\lambda \in \{5, \ldots, 15\}$ in steps of one. This effectively integrates this hyperparameter as it covers the entire empirically viable range. All training runs were carried out on a single NVIDIA RTX A6000 GPU and concluded in at most 1.5 hours. Mini-batching could further decrease this modest training time.

**Datasets.** We evaluate GC detection with GC-xLSTM on six diverse datasets. Obtaining objective truth about the underlying graph for real-world scenarios is a constant challenge in Granger causality research. We thus employ the Lorenz-96 system of differential equations [Karimi and Paul, 2010], realistic fMRI brain activity simulations [Smith et al., 2011], and simulated linear VAR data following Tank et al. [2022]. For further qualitative insights on real-world data, we additionally analyze the Moléne weather dataset [Girault, 2015], human motion capture recordings [CMU, 2009], and company fundamentals [Divo et al., 2025]. An overview and more details are provided in Appendix C.

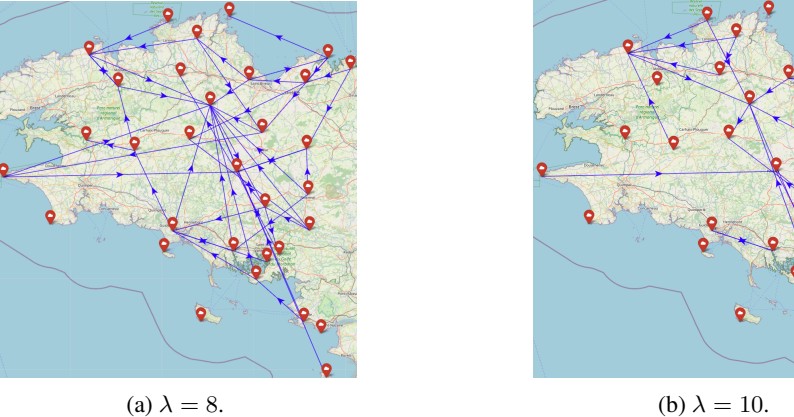

(a) $\lambda = 8$.  (b) $\lambda = 10$.

Figure 3: **GC-xLSTM uncovers dynamic GC weather patterns in the Moléne dataset.** We observe that the sparsity of the learned Granger causal relations increases with higher $\lambda$.

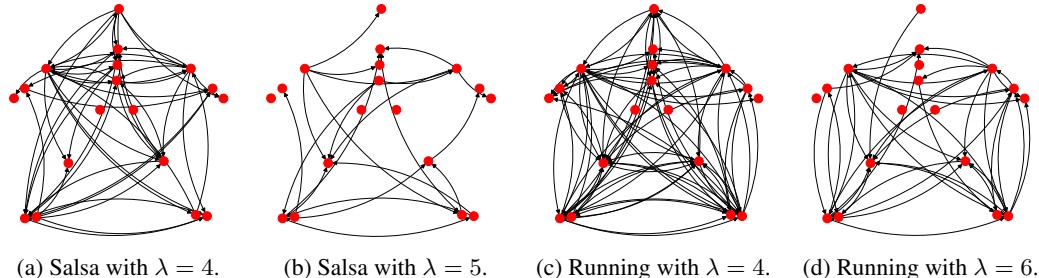

(a) Salsa with $\lambda = 4$.  (b) Salsa with $\lambda = 5$.  (c) Running with $\lambda = 4$.  (d) Running with $\lambda = 6$.

Figure 4: **GC-xLSTM captures complex human motions.** GC-xLSTM is able to uncover complex real-world dependencies in the Human Motion Capture dataset, giving us an intuitive understanding of the learned interactions.

### 4.1 Main Results

**Lorenz-96.** Table 1 evaluates GC detection with GC-xLSTM on three metrics. As a qualitative comparison, we provide results for seven well-known baselines: VAR as classic $F$-tests for Granger causality taken from Marcinkevičs and Vogt [2021], cMLP and cLSTM [Tank et al., 2022], GC-KAN [Lin et al., 2024], TCDF [Nauta et al., 2019], eSRU [Khanna and Tan, 2020], and GVAR [Marcinkevičs and Vogt, 2021]. Their scores are taken as reported in the original papers. First, GC-xLSTM outperforms all baseline methods for both $F = 10$ and $F = 40$ in both accuracy and balanced accuracy. This shows that GC-xLSTM reliably captures the underlying Granger causal relationships in the presence of limited and noisy data. Second,

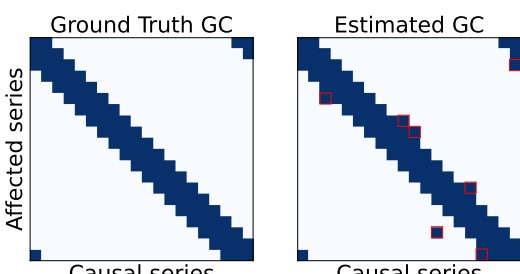

Figure 5: **GC-xLSTM uncovers the vast majority of GC edges.** In the highly chaotic $F = 40$ setting of the Lorenz-96 system GC-xLSTM is accurate in predicting the GC edges, shown in dark blue ■. Errors are marked red □.

it demonstrates that the sparsity hyperparameter $\lambda$ is rather well-behaved, as the AUROC score resulting from a sweep over a range of $\lambda$ values still provides very competitive results. Third, as a qualitative validation, Figure 5 visually confirms the strong prediction accuracies of GC-xLSTM.

**Moléne.** Unlike Zoroddu et al. [2024], who incorporate graph prior knowledge based on sensor locations, our approach learns the GC structure solely from the temperature observations. This ensures that GC-xLSTM does not inherently favor regional connections over long-range dependencies,

allowing it to discover dominant weather patterns operating both locally and across broader spatial scales. Adjusting $\lambda$ allows balancing granularity and interpretability for insights into both local and regional dependencies. The dense structure of the resulting Figure 3a exhibits a richer set of GC interactions, while the more sparse Figure 3b highlights only the most pronounced edges.

**fMRI.** Next, we evaluated the efficacy of GC-xLSTM in noisy settings by considering the rich and realistic BOLD deconvolved time series provided by the fMRI data. As the balanced accuracies (BA) in Table 2 show, GC-xLSTM significantly outperforms the baseline models.

Table 2: **GC-xLSTM discovers brain connectivity highly accurately.**

| Model | BA ($\uparrow$) |
| --- | --- |
| TCDF | 72.8±6.3 |
| GVAR | 65.2±4.5 |
| VAR | 51.3±1.5 |
| cMLP | 61.4±6.8 |
| cLSTM | 65.5±5.3 |
| GC-xLSTM | **73.3**±3.0 |

**Human Motion Capture.** To analyze GC relations between body joints, we focus on two specific activities: Salsa dancing and running, which provide interpretable motion patterns. Figure 4 shows the results of the learned graphs for those activities. A closer look offers an intuitive understanding of the learned interactions. For example, in the Salsa dance, we observe edges from the feet to the knees and the arms, supporting the characteristic movements of the lower driving the upper body. We can also see the cross-limb correlation, with movements initiating on one side of the body and propagating across. Similarly, the results for running strongly establish the lower limbs as primary motion drivers, with edges from the feet and knees to the arms. The cyclic dependencies between the knees, ankles, and feet capture the repetitive, alternating nature of the gait.

**Company Fundamentals.** Lastly, we evaluated how well GC-xLSTM can uncover relationships between financial indicators of large companies. The dataset consists of short time series of quarterly performance indicators of 2527 large publicly traded companies. As the excerpt in Figure 6 shows and as verified by a financial expert, those extracted edges are mostly economically sensible. Results on the full set of 19 features are provided in Appendix D.

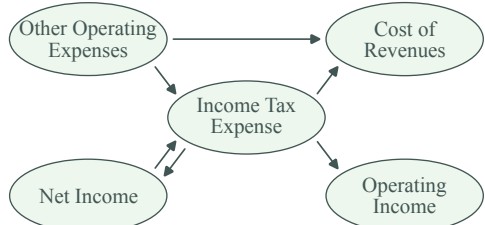

Figure 6: **GC-xLSTM extracts relations between company fundamentals.**

### 4.2 Model Analysis

**Ablation Study.** GC-xLSTM comprises two key innovations: Employing the xLSTM architecture (cf. Section 3.1) and a novel joint optimization strategy (cf. Section 3.2). To disentangle the empirical impact of each, we performed an ablation study. Table 3 to the right shows balanced accuracy results on the Lorentz ($F = 40$) and fMRI datasets. First, to assess the architectural contribution, we replaced the xLSTM block with a standard LSTM. The resulting substantial performance drop (GC-xLSTM $\rightarrow$ (I)) validates the importance of the xLSTM's advanced modeling capacity. Second, to evaluate the optimization strategy, we substituted it with standard Group Lasso [Simon and Tibshirani, 2012]. This also led to a marked decline in performance (GC-xLSTM $\rightarrow$ (II)), confirming its ability to enforce strict input sparsity.

Table 3: **Both the xLSTM forecaster and the novel joint optimization of GC-xLSTM drive its improved performance.**

| Ablation | Forecaster | Optimization | Lorentz | fMRI |
| --- | --- | --- | --- | --- |
| GC-xLSTM | xLSTM | Joint | **96.6**±0.3 | **73.3**±3.0 |
| (I) | LSTM | Joint | 93.0±0.3 | 62.8±2.0 |
| (II) | xLSTM | Group Lasso | 73.0±4.6 | 65.4±2.0 |

**Variable Number of Lags.** GC-xLSTM can naturally be extended to learn separate projections $\boldsymbol{W}^{(\ell)}$ per time lag $\ell$, effectively inflating $\boldsymbol{W}$ to rank three. We evaluate it on the simulated VAR dataset across different time series lengths $T$ and number of variates $V$ in Table 4, proving its ability to learn multiple lag-specific relations without training additional xLSTM models.

Table 4: **GC-xLSTM discovers specific relations per lag.** Balanced Accuracies for simulated VAR at different lengths $T$ and number of variates $V$.

| BA ($\uparrow$) | $T = 250$ | $T = 500$ | $T = 1000$ |
| --- | --- | --- | --- |
| $V = 10$ | 93.1±3.0 | 92.5±1.0 | 95.5±1.0 |
| $V = 20$ | 83.9±1.0 | 89.1±1.5 | 88.5±2.0 |

**Complexity and Scaling Behaviour.** To discover the entire GC graph, we need to fit $V$ models $\mathcal{M}_{\boldsymbol{\theta}_v,v}$ to obtain the respective sparse projection matrices $\boldsymbol{W}_v$ containing all edges arriving at each $v$. Assuming for brevity that the latent dimension $D \propto V$, i.e., is a fixed multiple of the number of variates $V$, each fitting runs in $O(TV^2/N_h)$ time and requires $O(TV^2/N_h)$ space, with $T$ being the time steps and $N_h$ the number of sLSTM heads. Their block-wise structure permits that, despite the squared effort, forward and backward passes are efficiently computable even for thousands of dimensions. Depending on whether all $V$ models are estimated in sequence or parallel, either the time or memory complexity multiplies by $V$ to arrive at the total cost. In practice, however, GC-xLSTM is extremely efficient on contemporary computing platforms due to the availability of highly optimized implementations for xLSTM layers. Figure 8 in Appendix E shows that it effectively scales linearly in the number of variates for the ranges relevant to standard GC detection settings.

**Inspecting Training Dynamics.** Here, we elaborate on the utility of the logarithm in the reduction loss $\mathcal{L}_{\text{red}}$ of Eq. (3). The logarithm term incentivizes the model to explore sparser solutions to GC discovery by allowing the training to move forward over any local minima that use the complete set of input variates. As the projection matrix becomes sparser and the input variates vanish from consideration by the model, the prediction loss $\mathcal{L}_{\text{pred}}$ will slightly increase. As seen in Figure 9 in Appendix F, an increase in the sparsity of the feature selectors $\boldsymbol{W}$ drives down the loss and enables learning more meaningful GC relations. It also shows how the variable usage quickly stabilizes.

## 5 Conclusion

We presented GC-xLSTM, a novel xLSTM-based model to uncover Granger causal relations from the underlying time series data. GC-xLSTM first enforces sparsity between the time series components and then learns a weight per time series to decide the importance of each time series for the underlying task. Each time series component is then modeled using a separate xLSTM model, which enables it to better discover Granger causal relationships between the time series variables. We validated GC-xLSTM in six scenarios, showing its effectiveness and adaptability in uncovering Granger causal relationships even in the presence of complex and noisy data.

**Limitations.** While Section 3.3 provides a theoretical analysis of GC-xLSTM, it does not give guarantees in the form of mathematical proofs. The rigor is limited by a lack of formal analysis of xLSTM blocks (see also Appendix B), which are yet to be formally analyzed to the same degree as more established architectures like LSTMs. While we discuss and measure the scaling behaviour of GC-xLSTM in Section 4.2, we only focused on dozens of variates as common in the literature.

**Future work.** This includes using more sophisticated architectures such as TimeMixer [Wang et al., 2024] or xLSTM-Mixer [Kraus et al., 2025]. Furthermore, discovering causal links specific to certain lags could be refined, where per-lag projections are learned for the near past and a remainder projection for the more distant lags. Finally, extending our evaluations to more real-world datasets encompassing domains such as climate change or ecology is an essential next step.

## Acknowledgments and Disclosure of Funding

This work received funding from the ACATIS Investment KVG mbH project "Temporal Machine Learning for Long-Term Value Investing" and the KompAKI project of the German Federal Ministry of Research, Technology and Space within the "The Future of Value Creation – Research on Production, Services and Work" program (funding number 02L19C150), managed by the Project Management Agency Karlsruhe (PTKA). The TU Eindhoven author received support from their Dep. of Mathematics and Computer Science and the Eindhoven AI Systems Institute. Furthermore, this work benefited from the DYNAMIC Centre funded by the LOEWE program of the Hessian Ministry of Science and Research, Arts and Culture (HMWK) as LOEWE 1/16/519/03/09.001 (0009)98 and the HMWK project "The Third Wave of Artificial Intelligence – 3AI". It also benefited from the early stage of the German Federal Ministry for Economic Affairs and Energy project "Souveräne KI für Europa" (13IPC040G) as part of the EU funding program IPCEI-CIS; funding has not started yet, and from early stages of the Cluster of Excellence "Reasonable AI" (EXC-3057) funded by the German Research Foundation (DFG) under Germany's Excellence Strategy; funding will begin in 2026. Funding for H. Poonia to attend NeurIPS was provided by the CMU GSA/Provost Conference Funding. The authors are responsible for the content of this publication. Map data © OpenStreetMap contributors, licensed under the ODbL and available from openstreetmap.org.

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

## A  Full Algorithm

Algorithm 1 describes the entire procedure for extracting a Granger causal graph using GC-xLSTM. $(\cdot)_+$ denotes truncation as $\max(\cdot, 0)$.

---

**Algorithm 1** Granger causality detection with GC-xLSTM

---

1: **Input:** Training data $\mathcal{D} = \left\{ \boldsymbol{S}^{(i)} \right\}_{i \in \{1, \ldots, N\}}$, sparsity hyperparameter $\lambda \in \mathbb{R}_+$, learning rate $\eta \in \mathbb{R}_+$, and compression schedule start $K \in \mathbb{N}$

2: **Output:** Granger causal graph

---

3:   **for all** $v \in \mathcal{V}$ **do**                                      ▷ Training decomposes over $\mathcal{V}$

4:       $\boldsymbol{\phi}_v : \boldsymbol{W}_v, \boldsymbol{b}_v \sim \mathcal{U}\left(-\sqrt{1/V}, \sqrt{1/V}\right)$             ▷ Kaiming/He initialization

5:       $\boldsymbol{\alpha}_v \leftarrow 1/V \mathbb{1}$                                ▷ Uniform reduction coefficient

6:       $\boldsymbol{\theta}_v \leftarrow \boldsymbol{\theta}^{(0)}$                              ▷ Standard xLSTM initialization

7:       $k \leftarrow 0$

8:       **repeat**

9:           Sample random mini-batch $\mathcal{B} \sim \mathcal{D}$

10:           $\boldsymbol{g}_{\phi, \alpha, \theta} \leftarrow \nabla_{\phi, \alpha, \theta} \frac{1}{|\mathcal{B}|} \sum_{\boldsymbol{S} \in \mathcal{B}} \left[ \mathcal{L}_{\text{pred}}(\boldsymbol{S}; \boldsymbol{\phi}_v, \boldsymbol{\theta}_v) + \lambda \log\left( \sum_{w=1}^{V} \alpha_v^w \left\| \text{sg}\left(\boldsymbol{W}_v^w\right) \right\|_2 \right) \right]$

11:           $\boldsymbol{\phi}_v \leftarrow \boldsymbol{\phi}_v - \eta \boldsymbol{g}_\phi$                          ▷ GD step

12:           $\boldsymbol{\theta}_v \leftarrow \boldsymbol{\theta}_v - \eta \boldsymbol{g}_\theta$                          ▷ GD step

13:           **if** $k \geq K$ **then**                ▷ Optimize $\boldsymbol{\alpha}$ after $K$ steps

14:              $\boldsymbol{\alpha}_v \leftarrow \boldsymbol{\alpha}_v - \eta \boldsymbol{g}_\alpha$                 ▷ GD step

15:           **end if**

16:           $\boldsymbol{W}_v \leftarrow \left( 1 - \frac{\lambda \eta \boldsymbol{\alpha}_v}{\|\boldsymbol{W}_v\|_2} \right)_+ \boldsymbol{W}_v$       ▷ Compression step with proximal GD

17:           $k \leftarrow k + 1$

18:       **until** convergence

19:   **end for**

20: $\mathcal{E} \leftarrow \left\{ (v, w) \in \mathcal{V} \times \mathcal{V} \mid \|\boldsymbol{W}_v^w\|_2 > 0 \right\}$

21: **return** extracted graph $(\mathcal{V}, \mathcal{E})$

---

## B  On the Approximation Capabilities of sLSTM blocks

Classic RNNs have long been known to be extremely powerful models of computation. Specifically, they are Turing complete [Siegelmann and Sontag, 1995] and, by extension, universal function approximators [Song et al., 2023]. Traditional LSTMs as proposed by Hochreiter and Schmidhuber [1997] are universal function approximators being at least as powerful as the RNNs [Song et al., 2023, Corollary 16]. Due to their novelty, the sLSTM *cells* as presented in Section 2.3 have not yet received the same degree of theoretical analysis.[2] Yet, it appears natural to extend the same reduction to RNNs as has been shown for LSTMs, since the main technicality is the differing normalization of the hidden state [Beck et al., 2024, Eq. 36] and the alternative exponential activation function, which Song et al. [2023] abstract away with. sLSTM *blocks* then wrap these cells with additional operations (cf. Beck et al. [2024], App. A.4). They, however, can be carefully configured to reduce to only rescaling Layer Normalizations and residual connections, which are known to, again, yield universal function approximators for many architectures [Tabuada and Gharesifard, 2020, Marchi et al., 2021]. This can be shown by first omitting the optional upfront convolution and Swish activation. We, furthermore, set the number of heads to $N_h = 1$, causing the per-head Group Normalization after the cell to degenerate to yet another Layer Normalization. Lastly, the final post up-projection of $\boldsymbol{x}_{\text{in}}$ to $\boldsymbol{x}_{\text{out}}$ is defined as

$$\boldsymbol{x}_{\text{out}} = \boldsymbol{W}_3 \left( (\boldsymbol{W}_1 \boldsymbol{x}_{\text{in}} + \boldsymbol{b}_1) \odot \text{GeLU}\left( \boldsymbol{W}_2 \boldsymbol{x}_{\text{in}} + \boldsymbol{b}_2 \right) \right) + \boldsymbol{b}_3.$$

---

[2]A discussion of expressivity hierarchies can be found in Auer et al. [2025, App. A].

With the right instantiation, where $\alpha$ is chosen such that $\text{GeLU}(\alpha) = 1$, we obtain

$$
\boldsymbol{x}_{\text{out}} = \begin{pmatrix} \mathbb{I} & 0 & 0 & 0 \\ 0 & \mathbb{I} & 0 & 0 \\ 0 & 0 & \mathbb{I} & 0 \end{pmatrix} \left( \left( \begin{pmatrix} \mathbb{I} & 0 & 0 \\ 0 & \mathbb{I} & 0 \\ 0 & 0 & \mathbb{I} \\ 0 & 0 & 0 \end{pmatrix} \boldsymbol{x}_{\text{in}} + \boldsymbol{0} \right) \odot \text{GeLU}\left( \boldsymbol{0}\boldsymbol{x}_{\text{in}} + \alpha\boldsymbol{1} \right) \right) + \boldsymbol{0}
$$

$$
= \begin{pmatrix} \mathbb{I} & 0 & 0 & 0 \\ 0 & \mathbb{I} & 0 & 0 \\ 0 & 0 & \mathbb{I} & 0 \end{pmatrix} \begin{pmatrix} \mathbb{I} & 0 & 0 \\ 0 & \mathbb{I} & 0 \\ 0 & 0 & \mathbb{I} \\ 0 & 0 & 0 \end{pmatrix} \boldsymbol{x}_{\text{in}} \odot \boldsymbol{1}
$$

$$
= \boldsymbol{x}_{\text{in}}.
$$

This, again, shows that sLSTM blocks are at least as general as LSTMs. While a rigorous proof is far beyond the scope of this work, this investigation still underpins the strong capabilities of this architecture. These theoretical considerations align well with empirical findings, showing that xLSTM blocks are at least as effective as LSTMs [Beck et al., 2024].

## C   Dataset Details

This section details all six datasets used in our empirical evaluation of GC-xLSTM. An overview is provided in Table 5.

Table 5: **Overview of the diverse collection of datasets.** It also shows the number of variates $V$, time steps $T$, samples $N$, and look-back context steps of GC-xLSTM $C$.

| Name | Origin | Type | $V$ | $T$ | $N$ | $C$ |
|---|---|---|---|---|---|---|
| Lorenz-96 | Karimi and Paul [2010] | Simulated | 20 | 500 | 1 | 10 |
| fMRI | Smith et al. [2011] | Real-world | 15 | 200 | 1 | 10 |
| Moléne | Girault [2015] | Real-world | 32 | 744 | 1 | 10 |
| Human MoCap (Run) | CMU [2009] | Real-world | 54 | 1232 | 61 | 10 |
| Human MoCap (Salsa) | CMU [2009] | Real-world | 54 | 4136 | 30 | 10 |
| Company Fundamentals | Divo et al. [2025] | Real-world | 19 | 56 | 2527 | 40 |
| VAR | Karimi and Paul [2010] | Simulated | 10/20 | mult. | 1 | 5 |

**Lorenz-96.**   The $V$-dimensional Lorenz-96 model [Karimi and Paul, 2010] is a chaotic multivariate dynamical system governed by the differential equations

$$
\frac{dx_{t,i}}{dt} = (x_{t,i+1} - x_{t,i-2})x_{t,i-1} - x_{t,i} + F,
$$

with the external forcing coefficient $F$ regulating the non-linearity of the system. Low values of $F$ correspond to near-linear dynamics, while higher values induce chaotic behavior. There are two sources of randomness in the system. Firstly, we sample i.i.d. starting conditions from $\mathcal{N}(0, 0.01)$. Secondly, in each step of the simulation, we add i.i.d. noise sampled from $\mathcal{N}(0, 0.1)$ to $x_{t,i}$. Following the setting of Tank et al. [2022] for best comparability, we simulate $V = 20$ variates with a sampling rate of $\Delta t = 0.05$ for a total of $T = 500$ time steps after a brief burn-in time. We use two forcing constants $F \in \{10, 40\}$ to test our model under different levels of non-linearity.

**fMRI.**   Discovering connectivity networks within (human) brains is a key application of GC detection methods. To this end, brain activity is measured non-invasively using functional magnetic resonance imaging (fMRI) over time and grouped into regions between which connections are looked for. Specifically, we used the realistic simulations of blood-oxygen-level-dependent (BOLD) deconvolved data of Smith et al. [2011].

**Moléne.**   The Moléne dataset [Girault, 2015] contains hourly temperatures recorded by sensors at $V = 32$ locations in Brittany, France, during $T = 744$ hours. The objective is to understand the spatio-temporal dynamics of the temperature and to assess the extent to which the model can uncover complex relationships in weather by considering only local observations.

**Human Motion Capture.** We also apply our methodology to detect complex, nonlinear dependencies in human motion capture (MoCap) recordings. In contrast to the Lorenz-96 simulated dataset results, this analysis allows us to visualize and interpret the learned network more easily. We consider a data set from the CMU MoCap database [CMU, 2009]. The data comprises $V = 54$ joint angle and body position recordings across multiple subjects. Since some regions, like the neck, have multiple degrees of freedom in both translation and rotation, we consider the GC relations between two joints based on edges between all movement directions. The motion ranges from locomotion (e.g., walking) over physical activities such as gymnastics and dancing to day-to-day social interactions.

**Company Fundamentals.** Another common field of application of GC detection methods is discovering links in economic data, where it is otherwise hard to maintain an overview. We, therefore, benchmark GC-xLSTM on company fundamentals data [Divo et al., 2025]. The dataset contains $V = 19$ economic variables, such as the Net Income and the Total Liabilities of 2527 companies. The data was collected quarterly from 2009 Q1 to 2023 Q3, resulting in only $T = 56$ time steps.

**VAR.** Following the well-known setup of Tank et al. [2022], we generate a two-step linear auto-regressive process in $V \in \{10, 20\}$ variates. All variables depend on themselves, and an additional three random dependencies are added as targets to be discovered. The number of time steps varies in $T \in \{250, 500, 1000\}$, and we start recording the data after a brief burn-in time.

## D   Extended Experimental Results

This section supplements the experimental findings of Section 4.1. Specifically, Figure 7 provides the complete set of relations extracted from the Company Fundamentals dataset. On this challenging full feature set, only some of the discovered GC relations are economically plausible.

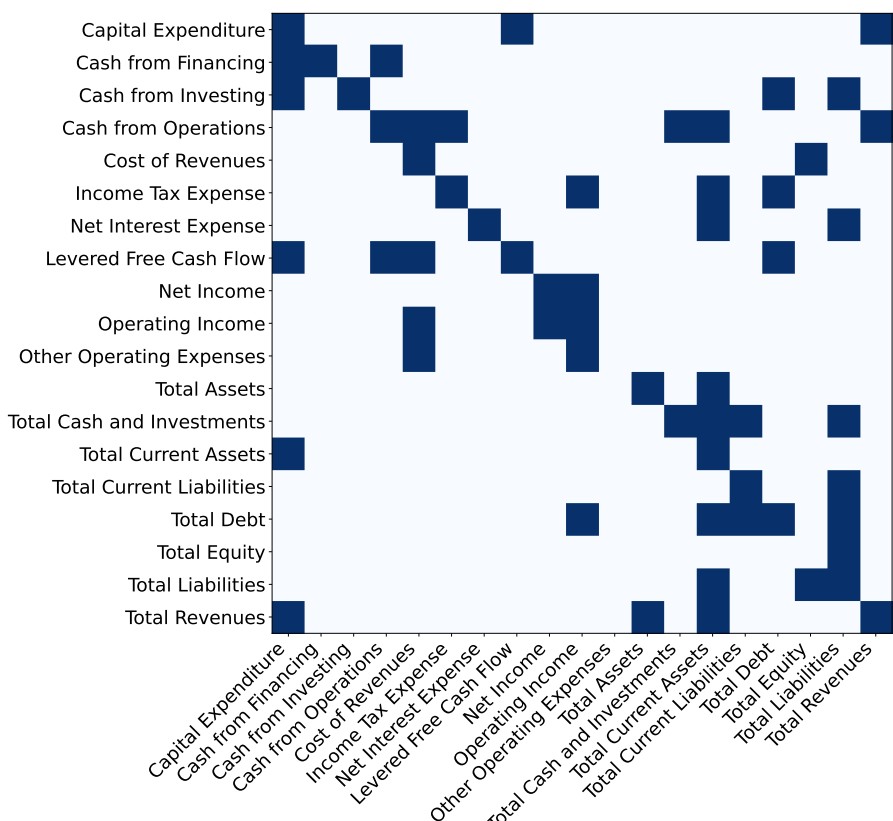

Figure 7: **GC-xLSTM uncovers relationships between economic variables within individual companies.** Established GC links are highlighted in **dark blue**.

# E  Scaling Behaviour

Figure 8 shows the training time and peak GPU memory reserved during the training of GC-xLSTM on the Lorenz-96 dataset with $T = 1000$ for various numbers of variates $V$.

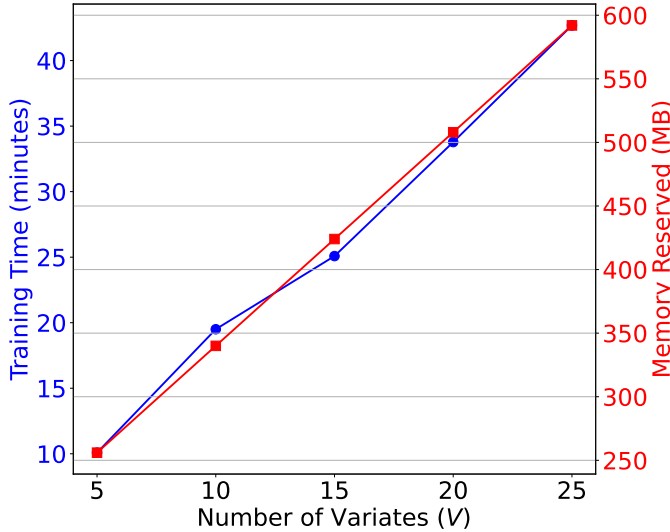

Figure 8: **GC-xLSTM effectively scales linearly in the relevant range of variate counts.**

# F  Additional Insights into Training

Figure 9a shows how the different loss components change during training. The robustness of training is reflected in the stability of variable usage once an optimal set is found, as Figure 9b shows.

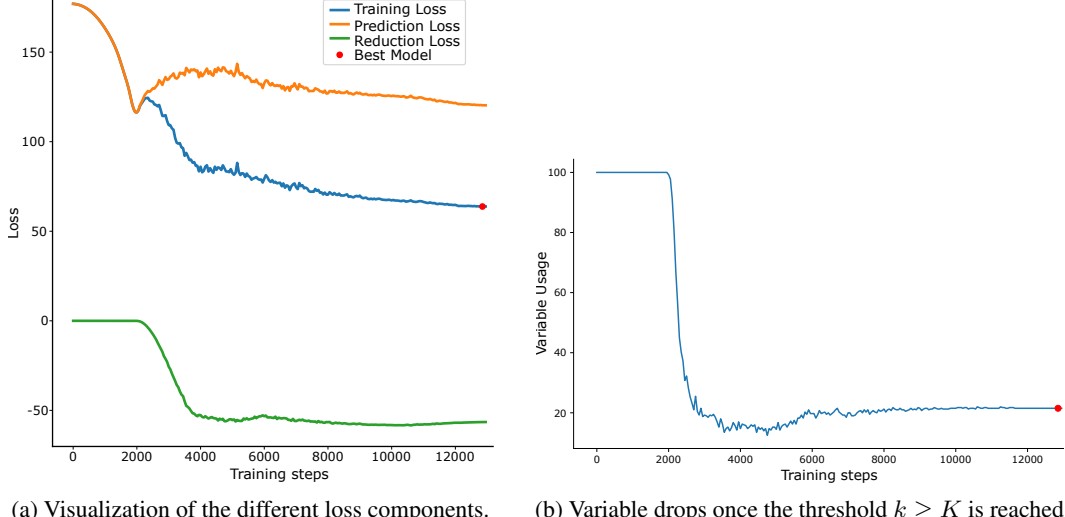

(a) Visualization of the different loss components.    (b) Variable drops once the threshold $k \geq K$ is reached and then quickly stabilizes.

Figure 9: **Algorithm 1 jointly optimizes the prediction loss while adaptively establishing sparsity.** Results show training on Lorenz-96 with $F = 40$ and $T = 1000$.

