# OpenReview forum: "Exploring Neural Granger Causality with xLSTMs: Unveiling Temporal Dependencies in Complex Data"
_NeurIPS.cc/2025/Conference — NeurIPS 2025 poster_

### Official Review · Reviewer_K18p · 2025-06-30

**Clarity:** 3
**Significance:** 2
**Originality:** 3
**Rating:** 4
**Confidence:** 5

**Summary:**

This paper develops Granger causal xLSTMs that use xLSTMS to learn granger causal relations from complex time series data with long time dependencies. The paper introduces a new model (GC-LSTM), formulates an optimization process for strict sparsity and provides evaluation of the model compared to benchmark models using six datasets.

**Questions:**

(Please see additional discussion related to these questions 1a and 1b in the Strengths & Weaknesses section).
1a. Why are other methods not sufficient for this problem?
1b. What is the significance of the GC-xLSTM model? Put another way, what is the particular advantage of this model? Does it better find long temporal dependencies? Can the authors provide evidence of this, and show that it has some major improvement over prior methods (as, with the current results it seems other methods actually work well.)

2. Appendix C – how many samples are in each dataset? The number of variables and timesteps are listed, but not the total number of samples.

3. I have additional concerns about scale, since this is a common issue with neural granger causality. Appendix F shows the scale with the variates V but how does the model scale over the number of timesteps T or the number of samples in the dataset?

**Ethical Concerns:**

["NO or VERY MINOR ethics concerns only"]

**Final Justification:**

The authors have addressed my core concerns in their rebuttal relating to significance and scalability. Therefore, I have raised my score from a 3 to a 4.

**Limitations:**

Yes

**Quality:**

3

**Strengths And Weaknesses:**

This paper has good quality and clarity: it is organized logically, explains the background extremely well, and describes the framework in a straightforward manner. The formulations seem technically sound and the model architecture and training seem logical. In terms of originality, the paper does propose a new model type of granger causal xLSTMs and includes an additional optimization and loss function focused on sparsity. The results show that the model is able to robustly find temporal dependencies from diverse data sources.

However, the point that is unclear for me is the significance: Why do we need this new model type? The introduction does a good job at defining granger causality, framing recent renewed interest in recurrent models and xLSTMs, and describing the combination of granger causality and xLSTMs, but what is missing here is a discussion about *why* these methods specifically should be combined, and why previous methods are not sufficient for learning long range dependencies in time series (and thereby justifying the need for this new model type).

This problem is further reiterated in the results section. I appreciate that the results section includes a wide range of results, including an ablation study, compares to many benchmarks, uses multiple datasets and provides some intuitive figures (e.g., Figure 4 in particular is very nice) showing the ability of the model to find underlying dependencies. However, these results do not convince me that this *specific* GC-xLSTM model is needed to solve the problem. For example, in Table 1 there is not a huge improvement compared to GVAR (and in some cases GVAR actually does better). Moreover, the figures (e.g., figures 3, 4, 5) do not compare with other models, making it unclear if this model learns dependencies any better than other methods.

Overall, I am willing to reevaluate if the authors can provide more compelling evidence about (a) why other methods are not sufficient and (b) show a major performance improvement with this model that justifies its need. For instance, if the authors are trying to make the claim that this method is better at finding long range dependencies between time series, from the current results it is not clear that the other methods do poorly for such dependencies – addition of other figures that might show much better found dependencies for the GC-xLSTM model compared to others would be helpful (e.g., like figures 3, 4 or 5), or additional performance tables (like Table 1 or 2) on the other datasets that show a large performance improvement across a range of tasks.

---

> ### Author Rebuttal · Authors · 2025-07-31
>
> Dear Reviewer K18p,
>
> Thank you for recognizing the work as clearly written and original.
>
> We gladly clarify the raised comments and questions:
>
> **Q1a:** Granger causality is not yet solved. For instance, looking at Tab. 2, we see that few methods even surpass just 70%, and all fall far short of the desired detection accuracies of 90+%. GC-xLSTM contributes toward reaching this by integrating xLSTM modules into a novel optimization procedure specifically designed to discover sparse inputs.
>
> **Q1b:** GC-xLSTM improves the detection accuracy of Granger causal graphs, as the varied empirical results shown specifically in Tab. 1 and 2 confirm. However, this does not yet show which of the two major contributions, (a) integrating xLSTMs and (b) the joint optimization procedure for sparse feature selection, contributes how much. We thus perform a respective ablation study on two datasets. Results are shown below.
>
> First, we replace the xLSTM block in the architecture with a standard LSTM layer, keeping the rest of the architecture unchanged. We can clearly see from (base) -> (a), that the performance drops substantially without the modeling capabilities of the xLSTM blocks.
>
> Second, we employ the baseline of Group Lasso (Simon and Tibshirani, 2012) instead of our novel procedure for optimizing for strict input sparsity. Again, we can see from (base) -> (b) that our strong results indeed stem from the contribution of the new optimization procedure. This new procedure is a contribution beyond GC-xLSTM and could, in the future, be used in other models too.
>
> Results on Lorenz-96 ($F=40$):
>
> | Model | Balanced Accuracy |
> |------------------------------|-----------------------------------|
> | (base) xLSTM + Joint Optimization   | 96.6% (±0.3) |
> | (a) LSTM + Joint Optimization    | 93.0% (±0.3) |
> | (b) xLSTM + Group Lasso | 75.0% (±4.0) |
>
> Results on fMRI:
>
> | Model | Balanced Accuracy |
> |------------------------------|-------------------------|
> | (base) xLSTM + Joint Optimization   | 73.3% (±3.0) |
> | (a) LSTM + Joint Optimization    | 62.8% (±2.0) |
> | (b) xLSTM + Group Lasso | 65.0% (±2.0) |
>
> **Q2:** An overview would help maintain a more complete image. We thus added the following table to Appendix C, amending the missing numbers:
>
> | Name | Origin | Type | Variates $V$ | Time Steps $T$ | Samples $N$ |
> |------|--------|------|---|---|---|
> | Lorenz-96 | Karimi and Paul [2010] | Simulated | 20 | 500 | 1 |
> | fMRI | Smith et al. [2011] | Real-world | 15 | 200 | 1 |
> | Moléne | Girault [2015] | Real-world | 32 | 744 | 1 |
> | Human MoCap (Run) | CMU [2009] | Real-world | 54 | 1232 | 61 |
> | Human MoCap (Salsa) | CMU [2009] | Real-world | 54 | 4136 | 30 |
> | Company Fundamentals | Divo et al. [2025] | Real-world | 20 | 56 | 2527 |
> | VAR | Karimi and Paul [2010] | Simulated | 10/20 | 250/500/1000 | 1 |
>
> **Q3:** We agree that scaling behavior is one of the key questions for any Granger causality method. Usually, though, the main challenge lies in scaling the number of variates $V$ (cf. ll. 97ff). We, therefore, specifically analyze it in Appendix F and already successfully apply the flexible GC-xLSTM framework to time series lengths between $T=56$ and $T=4136$ (see table directly above). Lastly, we also consider a wide range of samples from $N=1$ to $N=2527$. Note that here larger numbers of variates $N$ can actually be beneficial, since it reduces the likelihood of overfitting on smaller datasets.
>
> Thank you again for your thoughtful review. We hope it answered all outstanding questions. Specifically, having provided additional results on the main concern (significance), we would greatly appreciate it if you considered raising your score to reflect this.
>
> Best
>
> The Authors
>
> **References:**
> - Simon, N., & Tibshirani, R. (2012). Standardization and the group lasso penalty. Statistica Sinica, 22(3), 983–1001. [https://doi.org/10.5705/ss.2011.075](https://doi.org/10.5705/ss.2011.075)

---

> > ### Comment · Reviewer_K18p · 2025-08-04
> >
> > Thank you for your detailed rebuttal. My questions have been answered and all concerns related to significance and scaling alleviated. I have raised my score.

---

### Official Review · Reviewer_TkdF · 2025-07-01

**Clarity:** 4
**Significance:** 2
**Originality:** 2
**Rating:** 4
**Confidence:** 4

**Summary:**

This paper improves the detection of Granger causality using recent advances in deep learning for time series. In particular, it uses xLSTMs, which have demonstrated the ability to track longer-horizon relationships than traditional LSTMs. The paper introduces a loss function consisting of a sparsity-promoting feature selection and the xLSTM reconstruction. The new architecture is tested against a variety of GC methods on real and simulated data, where the method demonstrates good performance.

**Questions:**

1. What is the effect of the xLSTM vs. the new sparsity penalty?

2. Do you have any practical guidance on how to choose $\lambda$? This would greatly improve the practicality of the method, as it seems quite sensitive.

**Ethical Concerns:**

["NO or VERY MINOR ethics concerns only"]

**Final Justification:**

I still believe the use of xLSTMs is not particularly novel, and don't think the theoretical results are very strong. However, the authors have effectively addressed my concerns about ablation studies, which have further clarified that the optimization method proposed is actually a rather large component of the empirical success. I therefore raise my score, with the suggestion that the authors further emphasize the optimization algorithm over the use of xLSTMs in a camera ready version.

**Limitations:**

Yes

**Quality:**

3

**Strengths And Weaknesses:**

## Strengths

**Clarity** The paper is very well-written and easy to understand.

**Performance** The proposed method performs better than baselines, which are largely state-of-the-art. This is occasionally by somewhat large margins in the case of cLSTMs and cMLPs.

## Weaknesses

**Theoretical Analysis** I don't find the theoretical analysis particularly convincing. In particular, the point of the paper is that cLSTMs/cMLPs/etc. are not sufficient for Granger causality in complex settings, but unless I'm mistaken, the theoretical analysis applies equally well to those other architectures.

**AUROC Metric** GC-xLSTM is never the best model in terms of the reported AUROC, but I'm not super sure this is a meaningful metric, anyways. If I understand correctly, it is computed by sweeping over $\lambda$ -- but because of the group LASSO procedure done here, it's not obvious that $\lambda = 5$ for xLSTM is similar to $\lambda = 5$ in some other model.

**Lack of Ablation** This paper contributes both xLSTM-based GC, and a novel group LASSO-based sparsity penalty -- but these are not properly ablated. Thus, we cannot properly understand to what extent performance gains are due to xLSTMs, due to the new sparsity penalty, or both.

**Novelty** The use of xLSTMs rather than LSTMs is practically relevant and useful for the community, but is not significantly novel.

## Minor Typos
- Figure 6 Caption: "extracs" should be "extracts".
- Line 313, Left Column: "effectively inflating W it to rank" shouldn't have "it".

---

> ### Author Rebuttal · Authors · 2025-07-31
>
> Dear Reviewer TkdF,
>
> Thank you for recognizing the work as well-written, as well as the diverse evaluation and superior performance.
>
> We gladly clarify the raised comments and questions:
>
> **W1:** One *strength* of the theoretical analysis is that parts of it, indeed, apply to other works as well. You are correct in that models such as cLSTM would also, in theory, be able to learn GC relationships, which we never question. We merely posit that due to the increased modeling capabilities of xLSTMs, more complex relationships can be learned (Beck et al., 2024). We validate this hypothesis with empirical results. We also note that a major contribution besides the xLSTM is the novel optimization scheme, going beyond cLSTM etc.
>
> **W2:** Comparing very different algorithms by “integrating out” their hyperparameter choices, such as the sparsity penalty $\lambda$, is, indeed, challenging. They are comparable if the range of HPs each is complete, i.e., that the range covers both extremes where the behaviour saturates. This is the case for our AUROC computation. Also, note that the scores for cLSTM and cMLP were obtained analogously to our work. However, particularly when omitting this metric, GC-xLSTM clearly outperforms all baselines (cf. Tab. 1 & 2).
>
> **W3/Q1:** We agree that an ablation study to more clearly isolate the empirical contribution of these two components would be very valuable. We thus perform an ablation study of (a) the use of xLSTMs and (b) the joint optimisation strategy. The results are provided for two datasets and shown below.
>
> First, we replace the xLSTM block in the architecture with a standard LSTM layer, keeping the rest of the architecture unchanged. We can clearly see from (base) -> (a), that the performance drops substantially without the modeling capabilities of the xLSTM blocks.
>
> Second, we employ the baseline of Group Lasso (Simon and Tibshirani, 2012) instead of our novel procedure for optimizing for strict input sparsity. Again, we can see from (base) -> (b) that our strong results indeed stem from the contribution of the new optimization procedure.
>
> Results on Lorenz-96 ($F=40$):
>
> | Model | Balanced Accuracy |
> |------------------------------|-----------------------------------|
> | (base) xLSTM + Joint Optimization   | 96.6% (±0.3) |
> | (a) LSTM + Joint Optimization    | 93.0% (±0.3) |
> | (b) xLSTM + Group Lasso | 75.0% (±4.0) |
>
> Results on fMRI:
>
> | Model | Balanced Accuracy |
> |------------------------------|-------------------------|
> | (base) xLSTM + Joint Optimization   | 73.3% (±3.0) |
> | (a) LSTM + Joint Optimization    | 62.8% (±2.0) |
> | (b) xLSTM + Group Lasso | 65.0% (±2.0) |
>
> **W4:** Beyond the innovative and effective use of xLSTMs, we also propose a novel optimization procedure (Sec. 3.2, Alg. 1, Code), which we discuss in detail and validate empirically.
>
> **Minor typos:** Thank you, they are now fixed.
>
> **Q2:** Choosing hyperparameters is a common challenge in machine learning. However, $\lambda$ is rather well-behaved in two regards. Firstly, due to the adaptive joint optimization of both the sparsity projection and the forecast xLSTM, in many cases, results are rather stable across choices of $\lambda$. This really shows with the rather high AUROC scores of 88.0 and 99.3, which are computed over sweeps of $\lambda$ (cf. ll. 244ff). Secondly, when $\lambda$ does have a large effect, we can often visually inspect the resulting graph and coordinate with domain experts to determine the desired level of sparsity. As Fig. 4 shows, choosing $\lambda$ in this setting is not about model performance, but instead about the desired connectedness of the resulting graph, and therefore, an inherent user preference much like the threshold in other Granger causal detection methods.
>
> Thank you again for your thoughtful review. We hope it answered all outstanding questions. Specifically, having clarified the questions regarding the theory, the AUROC metric, and the hyperparameter choice, as well as providing the suggested ablation study, we would greatly appreciate it if you considered raising your score to reflect this.
>
> Best
>
> The Authors
>
> **References:**
> - Maximilian Beck, Korbinian Pöppel, Markus Spanring, Andreas Auer, Oleksandra Prudnikova, Michael K. Kopp, Günter Klambauer, Johannes Brandstetter, and Sepp Hochreiter. xLSTM: Extended Long Short-Term Memory. In The Thirty-eighth Annual Conference on Neural Information Processing Systems, November 2024.
> - Simon, N., & Tibshirani, R. (2012). Standardization and the group lasso penalty. Statistica Sinica, 22(3), 983–1001. [https://doi.org/10.5705/ss.2011.075](https://doi.org/10.5705/ss.2011.075)

---

> > ### Comment · Reviewer_TkdF · 2025-08-03
> >
> > Thanks for the responses. I include some further responses below.
> >
> > > Comparing very different algorithms by “integrating out” their hyperparameter choices, such as the sparsity penalty  , is, indeed, challenging. They are comparable if the range of HPs each is complete, i.e., that the range covers both extremes where the behaviour saturates. This is the case for our AUROC computation.
> >
> > I'm still not totally convinced here. In particular, AUROCs being comparable requires not only a similar range of extremes, but also a limited amount of nonlinearity in the relationship between the two quantities.
> >
> > > We thus perform an ablation study of (a) the use of xLSTMs and (b) the joint optimisation strategy. The results are provided for two datasets and shown below.
> >
> > These are very strong results! How are the hyperparameters chosen for Group Lasso? These results also suggest to me that the optimization procedure should perhaps be more central to the paper's narrative than xLSTMs.

---

> ### Author Response · Authors · 2025-08-04
>
> Thanks for engaging with the rebuttal.
>
> * Regarding the AUC, we agree that a bit of non-linearity can be important, and that is why the metric of choice is the balanced accuracy. The AUC metric was included for completeness reasons.
>
> * To choose the hyperparameters for group lasso, we do a sweep over lambda values. We also introduce a separate validation set for picking the best weights to prevent overfitting.
>
> We agree that the optimization procedure is central to our approach, and thus we highlight it as a major contribution (points 2 and 3 in key contributions on p. 2). Furthermore, please take a look at Fig. 8a in Appendix E (p. 16) that demonstrates the ability of our model to self-select a version of weights that gives the best GC accuracy.

---

> > ### Author Response · Authors · 2025-08-05
> > **Any further concerns?**
> >
> > Dear reviewer,
> >
> > As the discussion phase is coming to an end soon, we will be happy to answer any further concerns that you have. Your review has been helpful for us in making the manuscript stronger. We would appreciate an adjustment in the score if no further concerns remain.
> >
> > Regards,
> >
> > The authors

---

> > > ### Author Response · Authors · 2025-08-08
> > > **Discussion phase ending**
> > >
> > > Dear reviewer,
> > >
> > > Thank you for your engagement so far. SInce the discussion ends at AoE today we would like to take this oppurtunity to ask if there are any outstanding concerns from your part. We believe we have answered everything but will be sure to answer if something comes up. Thanks for the discussion till now.
> > >
> > > Regards,
> > >
> > > The Authors

---

### Official Review · Reviewer_dBaa · 2025-07-02

**Clarity:** 3
**Significance:** 2
**Originality:** 2
**Rating:** 5
**Confidence:** 4

**Summary:**

This paper introduces GC-xLSTM, a model to discover Granger causal relations from complex time series data, with a focus on long-range dependencies.
The method is based on a three step procedure. First, a sparse feature selector projects the input variables onto an embedding space using a sparse projection, that enables understanding what variables are relevant for predicting the future for each variable. Second, this time series of embeddings is fed to a xLSTM model trained to predict the next token. Finally, the coefficients of the sparse projection can be analyzed to reveal Granger causality. The whole method is trained end-to-end using a joint optimization technique that simulateously learns the parameters of the model and enforces sparsity.
The authors evaluate their method of diverse time series causality tasks such as Lorenz96 and fMRI datasets, showing it outperforms previous baselines.

**Questions:**

- I would like the experiments to demonstrate the added value of the optimization strategy proposed by the authors. Could the authors have an ablation of their model where they use common regularization on W and $\alpha$ and binarize entries using a threshold $\tau$ ? In my sense, this would validate the added value of the optimization strategy.
- Could the authors demonstrate more directly the impact of modeling long term dependencies in practice ?
- Similarly, the same method could be applied to any auto-regressive model. Could the authors show that using xLSTM rather than another autoregressive approach is beneficial ?

**Ethical Concerns:**

["NO or VERY MINOR ethics concerns only"]

**Final Justification:**

The reviewers have addressed my concerns

**Limitations:**

yes

**Quality:**

3

**Strengths And Weaknesses:**

- The paper tackles an important problem of inferring causality from time series. It is well written and proposes an elegant solution to the problem.
- The authors propose an effective optimization scheme to jointly train their model while enforcing strict sparsity in the coefficients.
- Despite the focus on long range  dependencies that may not be captured by other models, I could not find much experimental evidence for this advantage in the paper.
- The theoretical section 3.3 is not really a theoretical section but rather a high-level motivation for their architecture. Although obvious from the architecture, the authors could consider formalizing the Granger causality inferred by their model from the sparse coefficients.
- The long term dependency feature is directly inherited by using xLSTM and does not represent a major contribution.

---

> ### Author Rebuttal · Authors · 2025-07-31
>
> Dear Reviewer dBaa,
>
> Thank you for recognizing the clear description of the novel optimization scheme and comprehensive evaluation on a diverse set of benchmarks.
>
> We gladly clarify the raised comments and questions:
>
> **Q1:** We agree that this is an important ablation and since added it to the paper for the datasets Lorenz-96 and fMRI. Specifically, we employ the baseline of Group Lasso (Simon and Tibshirani, 2012) instead of our novel procedure for optimizing for strict input sparsity. We can see from (base) -> (ablated) that our strong results indeed stem from the contribution of the novel optimization procedure. It provides substantial and consistent improvements over the prior Group Lasso procedure.
>
> Results on Lorenz-96 ($F=40$):
>
> | Model | Balanced Accuracy |
> |------------------------------|-----------------------------------|
> | (base) xLSTM + Joint Optimization   | 96.6% (±0.3) |
> | (ablated) xLSTM + Group Lasso | 73.0% (±4.6) |
>
> Results on fMRI:
>
> | Model | Balanced Accuracy |
> |------------------------------|-------------------------|
> | (base) xLSTM + Joint Optimization   | 73.3% (±3.0) |
> | (ablated) xLSTM + Group Lasso | 65.4% (±2.0) |
>
> **Q2:** The motivation for employing xLSTM-based models is more than just capturing long-term dependencies, and also their generally improved modeling accuracy while still limiting the compute necessary (see, e.g., Beck et al., 2024). This is also confirmed empirically by the effectiveness of GC-xLSTM. Specifically, for the Company Fundamentals dataset, the model uses 40 time steps (10 years) instead of 10 steps (2.5 years), which we deem as already a long-term span in (Granger) causal discovery. See also the next paragraph for empirical proof of the benefit of using xLSTM modules over the classic LSTMs.
>
> **Q3:** To specifically show the benefit of this improved modeling capability, we ablated GC-xLSTM by replacing the xLSTM with a classic LSTM. The following table shows the results of replacing the xLSTM block in the architecture with a standard LSTM layer, keeping the rest of the architecture unchanged. We can clearly see from (base) -> (ablated), that the performance drops substantially without the modeling capabilities of the xLSTM blocks, especially in the case of the real-world data set of fMRI.
>
> Results on Lorenz-96 ($F=40$):
>
> | Model | Balanced Accuracy |
> |------------------------------|-----------------------------------|
> | (base) xLSTM + Joint Optimization   | 96.6% (±0.3) |
> | (ablated) LSTM + Joint Optimization    | 93.0% (±0.3) |
>
> Results on fMRI:
>
> | Model | Balanced Accuracy |
> |------------------------------|-------------------------|
> | (base) xLSTM + Joint Optimization   | 73.3% (±3.0) |
> | (ablated) LSTM + Joint Optimization    | 62.8% (±2.0) |
>
> Thank you again for your thoughtful review. We hope it answered all outstanding questions. Specifically, having clarified the questions regarding the long-term dependencies, as well as providing the suggested ablation study, we would greatly appreciate it if you considered raising your score to reflect this.
>
>
> Best
>
> The Authors
>
> **References:**
> - Simon, N., & Tibshirani, R. (2012). Standardization and the group lasso penalty. Statistica Sinica, 22(3), 983–1001. [https://doi.org/10.5705/ss.2011.075](https://doi.org/10.5705/ss.2011.075)

---

> ### Comment · Reviewer_dBaa · 2025-08-07
> **Thank you for your answers**
>
> I want to thank the reviewers for their thorough review. In particular, I appreciate the the additional experiments that, in my sense, provide enough evidence for the value of the xLSTM and the joint optimization. I encourage the authors to include it in their final manuscript. I'm updating my score accordingly.

---

### Official Review · Reviewer_w7eG · 2025-07-02

**Clarity:** 3
**Significance:** 2
**Originality:** 2
**Rating:** 4
**Confidence:** 3

**Summary:**

The paper proposes Granger causal xLSTMs (GC-xLSTM), which leverage the recent xLSTM architecture to capture long-range relations between variables. GC-xLSTM enforces sparsity between the time series components. The authors test the method on six diverse simulated and real-world datasets.

**Questions:**

I wonder how recent deep state space models perform. Some examples include [1-2], which has excelled in long-sequence modeling. I would like to get a justification for using xLSTM over other deep state space models.

[1] Smith, Jimmy TH, Andrew Warrington, and Scott W. Linderman. "Simplified state space layers for sequence modeling." arXiv preprint arXiv:2208.04933 (2022).

[2] Gu, Albert, and Tri Dao. "Mamba: Linear-time sequence modeling with selective state spaces." arXiv preprint arXiv:2312.00752 (2023).

In the architecture details paragraph, it says the linear layer predicts the next 10 steps from the preceding 10 timesteps. This number seems quite small compared to some of the long-range sequencing tasks that are used to evaluate modern long-range sequence models.

In the Main Results section, I am not fully convinced that GC-xLSTM outperforms other baseline models or provides an intuitive understanding.

**Ethical Concerns:**

["NO or VERY MINOR ethics concerns only"]

**Final Justification:**

Through rebuttal, the authors answered my concerns on significance and originality. I thus decided to increase my score.

**Limitations:**

Yes

**Paper Formatting Concerns:**

I did not notice any major formatting issues in this paper.

**Quality:**

3

**Strengths And Weaknesses:**

Quality: The submission is technically sound, with appropriate explanations in the background and methods sections.

Clarity: The paper is clearly written and well-organized, with informative background and methods sections. I especially liked the paragraph on explaining the intuition behind the gradient update step.

Significance: Although the paper is well-structured, I do not think the paper goes beyond the current neural Granger causality methods to be impactful for the community. I think the contribution of the paper is lacking.

Originality: Although the paper explores how the recently developed xLSTM architecture performs in recovering the Granger causal relations,  I do not think that the work provides sufficiently new insights, deepens understanding, or highlights important properties of existing methods.

---

> ### Author Rebuttal · Authors · 2025-07-31
>
> Dear Reviewer w7eG,
>
> Thank you for recognizing the work as well-written and technically sound, as well as the diverse evaluation of the method and baselines.
>
> We gladly clarify the raised comments and questions:
>
> **Regarding Significance and Originality:** We want to point out that the contribution of this work is twofold: While the exploration of the xLSTM architecture of GC detection is one part, as identified by the reviewer, the other part is the novel optimization scheme introduced and analyzed in detail in Sec. 3.2. It is summarized in Algorithm 1 and implemented in the provided code. The latter is a contribution beyond just xLSTMs.
> To more clearly isolate the empirical contribution of these two components, we perform ablation studies of (a) the use of xLSTMs and (b) the joint optimisation strategy. The results are provided for two datasets and shown below.
>
> First, we replace the xLSTM block in the architecture with a standard LSTM layer, keeping the rest of the architecture unchanged. We can clearly see from (base) -> (a), that the performance drops substantially without the modeling capabilities of the xLSTM blocks.
>
> Second, we employ the baseline of Group Lasso (Simon and Tibshirani, 2012) instead of our novel procedure for optimizing for strict input sparsity. Again, we can see from (base) -> (b) that our strong results indeed stem from the contribution of the new optimization procedure.
>
> Results on Lorenz-96 ($F=40$):
>
> | Model | Balanced Accuracy |
> |------------------------------|-----------------------------------|
> | (base) xLSTM + Joint Optimization   | 96.6% (±0.3) |
> | (a) LSTM + Joint Optimization    | 93.0% (±0.3) |
> | (b) xLSTM + Group Lasso | 73.0% (±4.6) |
>
> Results on fMRI:
>
> | Model | Balanced Accuracy |
> |------------------------------|-------------------------|
> | (base) xLSTM + Joint Optimization   | 73.3% (±3.0) |
> | (a) LSTM + Joint Optimization    | 62.8% (±2.0) |
> | (b) xLSTM + Group Lasso | 65.4% (±2.0) |
>
> **Q1:** When choosing model families, there often is a trade-off between efficiency and fidelity. Indeed, state space models (SSMs) make for highly efficient time series models. However, xLSTM-based models tend to generally provide better modeling accuracy while still limiting the compute necessary (see, e.g., Beck et al., 2024). This is also confirmed empirically by the effectiveness of GC-xLSTM. We will ensure to make this discussion more explicit and balanced in the final version of the paper, where § ll. 30ff currently only discusses why no Transformers have been employed.
>
> **Q2:** We first note that the xLSTM is a recurrent model, where forecasting 10 steps into the future auto-regressively is quite typical. While a look-back of 10 steps is indeed little compared to many long-term time-series forecasting methods, it is a lot in the regime of (Granger) causal discovery we contribute to. For instance, the seminal work of Tank et al. (2022) introducing Neural Granger Causality mostly uses a lookback of 2 or 5. Marcinkevičs and Vogt (2021) use values of 1 or 5. We did not scale beyond that, as higher numbers were not necessary to sufficiently forecast the datasets we investigated. Also, not that this regime is beneficial in many real-world datasets, where limits in length do not allow for substantially larger look-backs or forecast horizons.
>
> **Q3:** Firstly, regarding outperforming other baselines: We quantitatively compare the results of GC-xLSTM to those of baselines on the Lorentz-96 dataset in Tab. 1 and on the fMRI dataset in Tab. 2. For Lorentz-96, GC-xLSTM always wins in both Accuracy and Balanced Accuracy while staying very competitive for AUROC. We further reiterate that this AUROC score is computed by sweeping over different sparsity configurations $\lambda$ ll. 244ff, essentially removing that hyperparameter. For fMRI, GC-xLSTM outperforms all five baselines.
>
> Thank you again for your thoughtful review. We hope it answered all outstanding questions. Specifically, having clarified the questions regarding significance and originality, as well as providing the suggested ablation study, we would greatly appreciate it if you considered raising your score to reflect this.
>
> Best
>
> The Authors
>
> **References:**
> - Tank, A., Covert, I., Foti, N., Shojaie, A., & Fox, E. B. (2022). Neural Granger causality. IEEE Transactions on Pattern Analysis and Machine Intelligence, 44(8), [https://doi.org/10.1109/TPAMI.2021.3065601](https://doi.org/10.1109/TPAMI.2021.3065601).
> - Marcinkevičs, R., & Vogt, J. E. (2021, January). Interpretable models for Granger causality using self-explaining neural networks. In International Conference on Learning Representations.
> - Simon, N., & Tibshirani, R. (2012). Standardization and the group lasso penalty. Statistica Sinica, 22(3), 983–1001. [https://doi.org/10.5705/ss.2011.075](https://doi.org/10.5705/ss.2011.075)
> - Beck, M., Pöppel, K., Spanring, M., Auer, A., Prudnikova, O., Kopp, M., Klambauer, G., Brandstetter, J., & Hochreiter, S. (2024). xLSTM: Extended long short-term memory. Advances in Neural Information Processing Systems, 37, 107547-107603.

---

> > ### Author Response · Authors · 2025-08-04
> > **Any further concerns?**
> >
> > Dear reviewer,
> >
> > Since the discussion period will end in a couple of days, we would like to ask if there are any further questions from your side. We have replied in detail to all your original concerns and hope that we have alleviated your concens. We would be happy to discuss further if necessary.
> >
> > Regards,
> >
> > The authors

---

> > ### Comment · Reviewer_w7eG · 2025-08-04
> >
> > I thank the authors for their detailed response. It helped clarify my questions. I will adjust the score accordingly.

---

> > > ### Comment · Reviewer_TkdF · 2025-08-08
> > >
> > > Thanks for the clarifications, I will adjust my score accordingly.

---

### Official Review · Reviewer_jaiu · 2025-07-05

**Clarity:** 3
**Significance:** 2
**Originality:** 2
**Rating:** 4
**Confidence:** 2

**Summary:**

This paper investigates the problem of learning Granger causality from time series with long-range dependencies. The authors propose an Extended Long Short-Term Memory (xLSTM) architecture, designed to enhance the model's ability to capture temporal dependencies relevant for Granger causal inference. Experimental results on both synthetic and real-world datasets, such as the Moléne dataset, demonstrate improved performance over existing methods.

**Questions:**

See Weaknesses.

**Ethical Concerns:**

["NO or VERY MINOR ethics concerns only"]

**Final Justification:**

Thank you for the authors’ response. Most of my concerns have been addressed, and I will raise my score accordingly. However, due to the relatively weak theoretical contribution of this work, I am not inclined to give it a high score.

**Limitations:**

See Weaknesses.

**Paper Formatting Concerns:**

NAN

**Quality:**

2

**Strengths And Weaknesses:**

Strengths:

1. The paper is well-organized and clearly written.

2. Experiments on real-world data validate the practical utility and efficiency of the proposed approach.

Weaknesses:

1. A central concern is the identifiability of Granger causality using GC-xLSTM. Although the authors acknowledge this issue in the conclusion, it remains a critical point. In the causal discovery community, theoretical guarantees for identifiability are often essential to justify empirical findings.

2. The set of baseline methods appears outdated. I recommend organizing the baseline algorithms in a table with details such as publication year and key differences, to clarify the novelty and time-effectiveness of the proposed method.

3. The claim that existing methods fail to capture interactions between time series requires more justification. Many neural network-based methods, especially with sufficient depth and capacity, can model complex temporal dependencies. Furthermore, it remains unclear how modeling interactions enhances the identifiability of Granger causality. A more detailed explanation or illustrative example would help clarify this point.

---

> ### Author Rebuttal · Authors · 2025-07-31
>
> Dear Reviewer jaiu,
>
> Thank you for recognizing the clear writing as well as the wide set of experiments on synthetic and real-world data.
>
> We gladly clarify the raised comments and questions:
>
> **W1:** In the causal discovery community, guarantees on identifiability are indeed cornerstones of many important works. However, we want to point out that (classic) causality and our focus, Granger causality, are different fields closely related in purpose, not methodology (see also ll. 26-29). Granger causality is a statistical test that does not yield Pearlian causal graphs and is centered on the prediction task. Thus, Granger causality can indeed be helpful in identifying potential causal relationships, but it doesn't necessarily imply true causal mechanisms in the classical sense. There has been some work on equating Granger causality with classical structural causality, such as by Bodik and Paluš (2024), where, however, the focus was on extreme events. Thus, we hope that the reviewer will appreciate that proving “identification” in case of Granger causality is an open, non-trivial problem and warrants a completely separate study.
>
> In GC, identifiability is *defined* (see Def. 1) to be given when we can correctly learn the underlying time-evolving process $g$. This is the case whenever we use models with sufficient capacity and flexibility (universal approximators), which we discuss being the case for GC-xLSTM in Sec. 3.3 and extend on in Appendix B. We further note that even the seminal work of Tank et al. (2022), having introduced Neural Granger Causality, recognizes this analysis as the limit of current theoretical guarantees in their conclusion section.
>
> We do agree that this is indeed important future work. However, it goes beyond this work and overarches the entire family of Neural Granger Causality models, such as cMLP, cLSTM, GC-KAN, and our GC-xLSTM. We thus mention it in ll. 338-341.
>
> **W2:** Adding the publication year to Tab. 1 is a very good suggestion. We thus added the following column and will reorder the rows to follow an ascending timeline.
>
> | Name | Year |
> |------|------|
> | VAR | 2007 |
> | cLSTM | 2022 |
> | cMLP | 2022 |
> | GC-KAN | 2024 |
> | TCDF | 2019 |
> | eSRU | 2020 |
> | GVAR | 2021 |
> | GC-xLSTM (ours) | (2025) |
>
> As can be seen, we compare against recent works, such as cLSTM, cMLP, and GC-KAN. We will extend the manuscript with a more detailed description of the methods and how they differ from xLSTM-Mixer.
>
> **W3:** This appears to be a misunderstanding. We never claim that models such as cMLP “fail to capture interactions”. Instead, we state that they “may not capture interactions between time series and external factors *as effectively as xLSTMs* [emphasis added]” in ll. 48f. This improved modeling capability was also shown by Beck et al. (2024).
> This modeling capability is important for uncovering the data-generating process with sufficient fidelity, as discussed in W1 and Sec. 3.3. The improvement is most clearly shown in the results of Tab. 2, where the benchmark is sufficiently challenging to differentiate the models based on that. There, methods with simpler models (specifically, GVAR, VAR, cMLP, and cLSTM) discover far fewer correct edges than more advanced models such as the attention-based TCDF and our GC-xLSTM (overall best).
>
> Thank you again for your thoughtful review. We hope it answered all outstanding questions. Specifically, having clarified the questions regarding identifiability, choice of baselines, and modeling complex temporal dependencies, we would greatly appreciate it if you considered raising your score to reflect this.
>
> Best
>
> The Authors
>
> **References:**
> - Tank, A., Covert, I., Foti, N., Shojaie, A., & Fox, E. B. (2022). Neural Granger causality. IEEE Transactions on Pattern Analysis and Machine Intelligence, 44(8), [https://doi.org/10.1109/TPAMI.2021.3065601](https://doi.org/10.1109/TPAMI.2021.3065601).
> - Beck, M., Pöppel, K., Spanring, M., Auer, A., Prudnikova, O., Kopp, M., Klambauer, G., Brandstetter, J., & Hochreiter, S. (2024). xLSTM: Extended long short-term memory. Advances in Neural Information Processing Systems, 37, 107547-107603.
> -Bodik, J., Paluš, M. & Pawlas, Z. Causality in extremes of time series. Extremes 27, 67–121 (2024). https://doi.org/10.1007/s10687-023-00479-5

---

### Note · Authors · 2025-08-12

We sincerely thank all reviewers and the AC for their time, constructive feedback, fruitful discussions, and the updated scores. Your insights have helped us improve our work's clarity, scope, and impact. We will incorporate all suggestions into the camera-ready version.

The key updates made during the rebuttal phase are:

- Clarifications on Granger causal identifiability using GCxLSTM
- Stronger motivation and empirical support for recurrence and xLSTM modules
- Clarifications on hyperparameter search and computing AUROC over $\lambda$ values
- Ablation study on the core components (xLSTM -> LSTM, our novel joint optimization -> group Lasso)
- Overview of the characteristics of the six diverse datasets

We are deeply grateful — your thoughtful input has made our paper even stronger.

Kind regards,

The Authors

---

### Decision · Program_Chairs · 2025-09-17

**Decision:**

Accept (poster)

**Comment:**

This work introduces a model to discover Granger causal relations from complex time series data, with a focus on long-range dependencies. The method is based on a three-step procedure. First, a sparse feature selector projects the input variables onto an embedding space using a sparse projection, which enables understanding what variables are relevant for predicting the future for each variable. Second, this time series of embeddings is fed to an xLSTM model trained to predict the next token. Finally, the coefficients of the sparse projection can be analyzed to reveal Granger causality. The whole method is trained end-to-end using a joint optimization technique that simultaneously learns the parameters of the model and enforces sparsity. The authors evaluate their method of diverse time series causality tasks such as Lorenz96 and fMRI datasets, showing it outperforms previous baselines.

All reviewers agree that this work is worth being accepted at NeurIPS, and I recommend accepting this paper, under the condition that the authors highlight better in camera camera-ready version what piece of their framework is driving the improvements they see e.g. sparsity, neural network architecture etc., to make the contribution clear.